# A Tale of Two Strawberries: Conventional and Organic Open-Field Production in California

**Leslie J. Verteramo Chiu * and Miguel I. Gomez**

Charles H. Dyson School of Applied Economics and Management, Cornell University, Ithaca, NY 14850, USA; mig7@cornell.edu
\* Correspondence: ljv9@cornell.edu

**Abstract:** Organic produce in general is perceived as environmentally superior to conventional produce. This perception is what partially drives some consumers to pay a price premium for organic food. To understand the environmental impact across various categories of both production systems, we performed a life cycle analysis on organic and conventionally produced strawberries in California, following input estimates from extension reports. This study found that organic strawberries performed worse than conventional strawberries in almost all environmental impact categories by unit of land and unit of production. Organic strawberries generate 46% more carbon footprint than conventional strawberries. One of the main environmental impact contributors of organic production is the effect of transportation of compost, manure, and other organic inputs, which are required in large volumes per ha. The contribution of input delivery to total carbon footprint per ha of organic strawberry production is 33%, and for conventional strawberry production the contribution is 8%. Post-harvest processing of strawberries is the activity in both production systems that contributes the most to total GWP per ha of production, up to 40% for organic and 60% for conventional strawberries.

**Keywords:** life cycle analysis; organic crop; organic strawberries; carbon footprint

## 1. Introduction

Strawberries are important fruits in terms of value of production in the US. Strawberry production in the US is valued at about USD 3.196 billion with an annual production of 1.4 million tons [1]. About 90 percent of the US strawberry production is concentrated in California, where strawberry production is ranked 4th in the state in terms of crop value.

Most strawberries in the US, and specifically in California, are produced using conventional, non-organic practices. According to the California Strawberry Commission (2019), about 13% of California's Central Coast strawberry acreage (the main strawberry-producing region in California, spanning between Los Angeles and San Francisco) utilize organic production. The yields per acre of organic strawberry production in California is estimated to be about 20% lower than conventionally produced strawberry, and the total cost of production per acre is about 5% higher for organically produced strawberry. The benefit to the organic producer is in the price premium, which can be up to 50% higher than the farmgate price of conventional strawberry [2]. Consumers' higher willingness to pay for organic food is the result of the perception that organic food is healthier and more sustainable for the environment [3,4]. The United States Department of Agriculture (USDA) organic standards require certain management and production practices for a produce to be certified organic, like avoiding some synthetic agrochemicals (The organic standards of the USDA are available at: https://www.ams.usda.gov/grades-standards/organic-standards, accessed on 10 June 2023) [5].

Organic production, by limiting the use of synthetic fertilizers, herbicides, and pesticides, is seen by some consumers as an ethical, sustainable, and healthier alternative.

A meta-analysis assessing the nutritional content of organic fruits and vegetables found that on average, organic fruits and vegetables are reported to have a higher nutritional content [6]. In a study comparing nutritional quality and pesticide residues between organic and conventional strawberries, the authors found no statistical differences in nutritional quality or antioxidant properties between the two production systems, including secondary metabolites [7]. Organic strawberries had no detectable pesticide residue and organic strawberries had pesticide residue below the limit of quantification. Although there have been studies on the environmental impact of organic vs. conventional agricultural products (see, for example, [8–10]), the results do not show, or cannot conclude, that organic production is categorically more sustainable than conventional food production. For instance, a systematic review [9] found 34 studies of agricultural products, mostly in Europe, where the relative difference in global warming potential (GWP) of organic fruits and vegetables production compared to conventional production ranged from −81 to 130% per product unit, indicating that at least one study (out of eight for this category) found that organic production has 130% higher GWP than conventional production. Looking at eutrophication potential of fruits and vegetables, another study [10] found the relative difference (organic vs. conventional) to range from −90 to 323%. For the specific case of strawberry production, two regions of California (South Coast for conventional, and Central Coast for organic) were compared in a study [10] using data from 2006, and the authors concluded that organic strawberries generate 30% less carbon footprint than conventional strawberries, without reporting any other environmental impact category.

Reporting only one impact category, GWP, in life cycle analysis (LCA) is a common practice. This single impact category reporting, however, limits the understanding of the overall environmental impact. A systematic review [11] reported that the number of environmental impact categories used in livestock life cycle analyses and found that 96% of the papers reported only one impact category, and 98% of the papers in their review reported GWP.

Consequently, the purpose of this study is to estimate the environmental impact of conventional and organic strawberry production in California. Life cycle analyses are conducted for the two production technologies located in the same production region. The life cycle inventory (LCI) was created from production cost analyses produced by the University of California Davis Extension Services [12,13]. We focus only on variable factors of production. Our analysis shows that organically produced strawberries are not categorically environmentally superior to non-organic produced strawberries, and that the results are sensitive to yields, i.e., they are stochastic.

## 2. Methodology

### 2.1. Scope of the Study

We estimated the environmental impact of conventional and organic strawberries produced in open fields in California's Central Valley. The LCA was conducted using SimaPro®, Release 9.0.0.49, following the method of ReCiPe 2016 Midpoint (H) v1.03/World (2010) H. The functional unit (FU) is a kg of harvested strawberries, and the system boundary is from cradle to farmgate. Environmental impacts per ha of production were estimated as well.

### 2.2. Data

The data obtained for the life cycle inventories for both organic and conventional production were obtained from the strawberry production and harvest cost analyses published by the UC Davis Agriculture and Natural Resources Cooperative Extension [12,13]. The cost analyses of the two production systems were estimated for the same region, specifically Santa Cruz and San Benito Counties for conventional production and Santa Cruz and Monterey Counties for organic production. The extension report of the cost estimates clarify that individual farms may have different cultural practices, but the report considers a typical well-managed farm in the region [12,13]. All input values were transformed to

the metric system and are available in Appendix A in Tables A1 and A2. The reports serve as guides for various types of strawberries without assuming a particular variety. The data are assumed to be for a representative farm of the region. Variable production inputs are included in the analysis without considering the effect of agricultural land transformation, civil infrastructure, or machinery manufacturing. The contribution of tertiary emissions, like machinery manufacturing and infrastructure, are not significant to total emissions in agriculture. A study [14] found that the contribution of tertiary emissions among 11 crops in nine countries in Europe was estimated at 3.3% of total field operation emissions. There are other LCA studies that do not consider tertiary emission [15–17]. The LCI for both production systems are estimated for the Central Coast Region of California. We assume that both production systems have the same environmental impact on energy demanded and water used.

The energy required to pump water for drip irrigation was estimated from a study [18], assuming a pump efficiency of 85%. For both production systems, irrigation water is assumed to be pumped from wells. A study [19] estimated the energy demanded and water used to sort and wash a ton of tomato, which they pointed out could be applied to similar crops. We included energy for sorting the produce, same for both production systems, at a rate of 1.43 kWh/ton; the water used for washing the produce is assumed to come from the city water supply at a rate of 380 L/ton [19].

In some instances, SimaPro® may not have the exact agricultural input in their libraries. When this was the case, we selected from the libraries that were a close substitute for the original input. For instance, we converted the amount of compost and feather meal in the original list of materials [13] to a manure equivalent since compost was not available in the libraries. We used the estimated nutrients from manure and compost from a study [20] to reach a quantity of manure with equivalent nutrition content as compost. For the specific case of some insecticides and fungicides, where the cost report only provided the commercial name of the input, the generic pesticide and fungicide were selected from the SimaPro® libraries.

The amount of plastic used in the drip tape, plastic mulch, and other plastic inputs, was calculated from the density of commercial product descriptions. We assumed the plastic to be polyethylene terephthalate (PET) with a usable life of 3 years. We included post-harvest plastic containers made of PET, at the rate of 26.6 kg/ton for both production systems [19,21].

A description of the production inputs used in the LCA in both production systems and their quantity used per ha are presented in Table 1.

**Table 1.** List of materials and inputs for each production system per ha.

| Production System | Materials | Amount | Unit |
| --- | --- | --- | --- |
| ***Conventional*** | | | |
| *Water and machinery* | Water (well) | 6985 | t |
| | Diesel | 683 | kg |
| | Gasoline | 311 | kg |
| | Polyethylene | 150 | kg |
| *Fertilizer* | | | |
| | N | 168 | kg |
| | $K_2O$ | 73 | kg |
| | $P_2O_5$ | 45 | kg |
| *Agrochemicals* | | | |
| | Fungicide | 51 | kg |
| | Pesticide | 10 | kg |
| *Post-harvest* | | | |
| | Tap water | 30,717 | kg |
| | PET | 2150 | kg |

**Table 1.** *Cont.*

| Production System | Materials | Amount | Unit |
|---|---|---|---|
| *Organic* | | | |
| *Water and machinery* | Water (well) | 6985 | t |
| | Diesel | 1132 | kg |
| | Gasoline | 550 | kg |
| | Polyethylene | 150 | kg |
| *Fertilizer* | | | |
| | Poultry manure | 6714 | kg |
| | Rice bran | 12,350 | kg |
| *Post-harvest* | | | |
| | Tap water | 23,862 | kg |
| | PET | 1672 | kg |

Rice bran is a byproduct of rice processing and is applied as an alternative to soil fumigation in the organic production system. It promotes anaerobic soil disinfestation, which is effective in controlling Verticillium wilt [22]. Rice bran was not included as an input in the UC Davis organic cost study of 2006, which was used as reference [10], while the use of other inputs remains similar. In general, the environmental contribution of rice bran, or any byproduct of a process, including manure, could be estimated by mass, which for rice bran is estimated to be about 10% of paddy rice [23]; by economic value, currently rice bran price is about 30% of the price of rice (https://www.ams.usda.gov/mnreports/ams_1655.pdf, accessed on 21 May 2023), or by gross energy allocation [24]. In some instances, the byproducts or the material that is being recycled from a process are treated as part of the production process and, under the cut-off system model, these materials are considered burden-free (https://ecoinvent.org/the-ecoinvent-database/system-models/, accessed on 21 May 2023). However, a market for byproducts warrants the use of an allocation method. In this analysis, we use the mass allocation model (10% of the environmental impact of rice production).

Some organic fertilizers were omitted from the analysis because no close substitute could be found. The amount of the excluded items is small (e.g., Biomin Calcium, Maxi-Crop, and Agrothirive LF) compared to the manure and rice bran volumes. We do not believe the effect on these minor organic inputs is significant to the LCA due to the small quantity used per ha. Our results correspond to a conservative estimate on the organic production system.

Transportation of organic fertilizer and rice bran is assumed to be 112 km [10,25], while transportation of synthetic fertilizer is assumed to be 200 km [10]. The plastics used in production and packing are also assumed to have an average transportation of 200 km. Once standardized to tkm (tkm is the product of mass transported (in tons) and km traveled) (tons per km), the contribution of plastics and fertilizers (other inputs) to organic production is 15% and 85%, respectively, of total transportation impact. The contribution of the same items is reversed for conventional production: plastics contribute to 86% of the transportation impact while agrochemicals contribute to 13% of total transportation impact. The total tkm for organic and conventional production are 2499.57 and 529.40, respectively.

Table 1 shows that the organic system consumes 66% more diesel and 77% more gasoline than the conventional system. The increase in hydrocarbon fuel may be the result of a substitution of agrochemical use for weeding and insect control, i.e., to a more manual control requiring more equipment use in the field. Tap water and PET use are directly related to the production volume.

## 3. Results

We present the LCA results of the environmental impact on several categories to produce a FU of organic and conventional strawberries, as well as the impact on ha of each production system.

### 3.1. Environmental Impact of Each Production System

The following table presents the impact on FU of strawberry under the mass allocation method for rice bran. Starting with GWP, conventional strawberries generate about eight percent more carbon footprint (in kg of $CO_2$ equivalent) than organic strawberries. This difference is much smaller than the estimated difference of 30% from a study [10]. The carbon footprint of agricultural produce depends on the level of technology and productivity, in other words, on the expected production per area. Given the stochastic nature of food production, the statistical significance of any environmental impact category per FU, given the same input level, is an empirical question, especially for those impact categories where the relative difference between the two production systems is small.

The results shown in Table 2 are based on the expected productions of 80,836 kg and 62,873 kg per ha of conventional and organic production, respectively [12,13]. For comparison, the expected production used in a study [10], obtained from extension reports [26,27] was 48,152 kg and 33,611 kg per ha of conventional and organic production, respectively. Productivity has increased by more than 60% for conventional and more than 85% for organic, in the last 15 years. Input use did not change much.

**Table 2.** Environmental impact per FU of both production systems.

| Impact Category | Unit | Org | Conv | % Diff. |
|---|---|---|---|---|
| Global warming | kg $CO_2$ eq | 0.22973 | 0.15671 | 46.60% |
| Stratospheric ozone depletion | kg CFC11 eq | $4.0 \times 10^{-7}$ | $6.0 \times 10^{-7}$ | −36.33% |
| Ionizing radiation | kBq Co-60 eq | 0.00960 | 0.00833 | 15.28% |
| Ozone formation, human health | kg NOx eq | 0.00066 | 0.00036 | 87.32% |
| Fine particulate matter formation | kg PM2.5 eq | 0.00039 | 0.00023 | 67.04% |
| Ozone formation, terrestrial ecosystems | kg NOx eq | 0.00069 | 0.00037 | 87.02% |
| Terrestrial acidification | kg $SO_2$ eq | 0.00112 | 0.00059 | 88.01% |
| Freshwater eutrophication | kg P eq | 0.00005 | 0.00004 | 36.01% |
| Marine eutrophication | kg N eq | 0.00008 | 0.00001 | 569.32% |
| Terrestrial ecotoxicity | kg 1,4-DCB | 0.79826 | 0.50550 | 57.92% |
| Freshwater ecotoxicity | kg 1,4-DCB | 0.00978 | 0.00395 | 147.31% |
| Marine ecotoxicity | kg 1,4-DCB | 0.01153 | 0.00567 | 103.34% |
| Human carcinogenic toxicity | kg 1,4-DCB | 0.00805 | 0.00537 | 49.92% |
| Human non-carcinogenic toxicity | kg 1,4-DCB | 0.23102 | 0.12692 | 82.02% |
| Mineral resource scarcity | kg Cu eq | 0.00057 | 0.00050 | 15.17% |
| Fossil resource scarcity | kg oil eq | 0.11205 | 0.07412 | 51.18% |
| Water consumption | $m^3$ | 0.11772 | 0.08889 | 32.43% |

Org: organic. Conv: conventional. FU: functional unit (1 kg). % diff: the percentage difference of organic with respect to conventional. Positive values indicate the impact of organic production is larger than the impact of conventional production.

The impact category where organic production fares the best is in stratospheric ozone depletion, with about 36% lower impact than conventional production. However, organic production has a marine eutrophication impact that is 569% higher than that of conventional production. The higher value of marine eutrophication impact of organic production is attributable to the use of manure that substitutes synthetic fertilizers. This soil nutrient enrichment creates changes in the marine ecosystem by promoting algae growth [28]. GWP per FU is 46% higher for organic strawberry. Venkat summarizes some points about the observed low environmental performance of organic production [10], which includes the low productivity with respect to conventional production, and conventional production generates more higher emissions per FU from energy use in the farm. Despite the low GWP of compost, the large amount of compost used in organic production adds up to total GWP per FU. Venkat shows that some organic crops can generate up to 480% more GWP (walnuts) per FU than conventional, with other crops like lettuce generating 40% more GWP than conventional production [10].

The following table presents the results per ha for the same impact categories.

The effects per ha of production show two changes in sign for the relative effects compared to the FU analysis. Overall, by looking at Tables 2 and 3, the same conclusion regarding comparative environmental impacts is reached. Reporting the environmental impact per area or production is relevant in understanding the effects of possible gradual production system changes over time or caused by policy changes, like restricting the use of some synthetic fertilizers.

**Table 3.** Environmental impact per ha of both production systems.

| Impact Category | Unit | Org | Conv | % Diff. |
|---|---|---|---|---|
| Global warming | kg $CO_2$ eq | 14,443.95 | 12,667.46 | 14.02% |
| Stratospheric ozone depletion | kg CFC11 eq | 0.022304 | 0.045037 | −50.48% |
| Ionizing radiation | kBq Co-60 eq | 603.847 | 673.469 | −10.34% |
| Ozone formation, human health | kg NOx eq | 41.809 | 28.697 | 45.69% |
| Fine particulate matter formation | kg PM2.5 eq | 24.431 | 18.805 | 29.92% |
| Ozone formation, terrestrial ecosystems | kg NOx eq | 43.363 | 29.811 | 45.46% |
| Terrestrial acidification | kg $SO_2$ eq | 70.114 | 47.948 | 46.23% |
| Freshwater eutrophication | kg P eq | 3.403 | 3.217 | 5.79% |
| Marine eutrophication | kg N eq | 4.876 | 0.937 | 420.58% |
| Terrestrial ecotoxicity | kg 1,4-DCB | 50,189.102 | 40,862.52 | 22.82% |
| Freshwater ecotoxicity | kg 1,4-DCB | 614.638 | 319.540 | 92.35% |
| Marine ecotoxicity | kg 1,4-DCB | 724.966 | 458.391 | 58.15% |
| Human carcinogenic toxicity | kg 1,4-DCB | 506.089 | 434.029 | 16.60% |
| Human non-carcinogenic toxicity | kg 1,4-DCB | 14,525.027 | 10,260.004 | 41.57% |
| Mineral resource scarcity | kg Cu eq | 35.960 | 40.144 | −10.42% |
| Fossil resource scarcity | kg oil eq | 7044.845 | 5991.176 | 17.59% |
| Water consumption | $m^3$ | 7401.369 | 7185.855 | 3.00% |

Org: organic. Conv: conventional. % diff: the percentage difference of organic with respect to conventional. Positive values indicate the impact of organic production is larger than the impact of conventional production.

Per unit of area, GWP per ha, of organic production is 14% higher than that of conventional production. Similarly, the impact of marine eutrophication in the organic system is 420% higher for organic production. What these values also indicate is the difference in production that one system needs to have over the other system to have equal environmental impact per FU. For instance, a GWP difference of 14% means that the GWP per FU of both systems is equal when conventional production increases by 14%.

The values presented in Tables 2 and 3 are conditional upon local conditions (soil quality, management, input use, external factors, and productivity). These conditions are not fixed across time and space; thus, the relative impact of either system may need to be evaluated regularly.

*3.2. Environmental Impact of Each Production Process and Input for Each Production System*

The relative impact of each production process and input on the impact categories analyzed are presented in the following Table 4. Each production system is divided into the following processes and inputs: diesel and gasoline, input delivery, polyethylene (field plastics), fertilizer, and post-harvest operations.

In the case of organic production, the impact of input delivery is 33% of total GWP, and in contrast, fertilizer use is 15% of total GWP contribution. This shows that the GWP of organic fertilizers (including rice bran) is not very large compared to the environmental impact of transporting these organic fertilizers due to their large volume needed per ha. Post-harvest does contribute substantially (41%) to the total GWP. Field plastics are not that important in terms of environmental impact (2% of contribution to total GWP). However, the effect of organic fertilizers on marine eutrophication is considerable at 94% of total effect. Input delivery and post-harvest processing are the main activities that negatively impact environmental performance (Table 5).

**Table 4.** Relative environmental impact per ha of the production processes and inputs for organic production.

| Impact Category | Diesel and Gasoline | Input Delivery | Field Plastics | Fertilizer | Post-Harvest |
| --- | --- | --- | --- | --- | --- |
| Global warming | 7.85% | 33.06% | 2.17% | 15.80% | 41.12% |
| Stratospheric ozone depletion | 8.33% | 12.43% | 0.06% | 73.05% | 6.13% |
| Ionizing radiation | 11.96% | 28.38% | 0.06% | 7.29% | 52.32% |
| Ozone formation, human health | 8.83% | 50.17% | 1.45% | 8.56% | 31.00% |
| Fine particulate matter formation | 12.44% | 30.66% | 1.10% | 20.51% | 35.29% |
| Ozone formation, terrestrial ecosystems | 9.02% | 49.89% | 1.51% | 8.41% | 31.16% |
| Terrestrial acidification | 12.44% | 22.90% | 1.07% | 36.46% | 27.14% |
| Freshwater eutrophication | 4.10% | 28.77% | 0.12% | 13.44% | 53.57% |
| Marine eutrophication | 1.77% | 1.36% | 0.01% | 94.46% | 2.40% |
| Terrestrial ecotoxicity | 3.25% | 49.49% | 0.28% | 6.03% | 40.95% |
| Freshwater ecotoxicity | 1.72% | 39.42% | 0.10% | 33.93% | 24.82% |
| Marine ecotoxicity | 2.45% | 48.21% | 0.13% | 18.93% | 30.28% |
| Human carcinogenic toxicity | 4.38% | 42.00% | 0.84% | 4.01% | 48.77% |
| Human non-carcinogenic toxicity | 2.46% | 58.11% | 0.12% | 6.83% | 32.48% |
| Mineral resource scarcity | 5.10% | 45.57% | 0.18% | 6.61% | 42.54% |
| Fossil resource scarcity | 29.65% | 22.56% | 3.34% | 4.21% | 40.24% |

Field plastics are all plastics used in the production process. Fertilizer includes manure and rice bran. Post-harvest includes washing, sorting, and packing.

**Table 5.** Relative environmental impact per ha of the production processes and inputs for conventional production.

| Impact Category | Diesel and Gasoline | Input Delivery | Field Plastics | Agrochemicals | Post-Harvest |
| --- | --- | --- | --- | --- | --- |
| Global warming | 5.26% | 7.98% | 2.47% | 24.00% | 60.29% |
| Stratospheric ozone depletion | 2.42% | 1.30% | 0.03% | 92.35% | 3.90% |
| Ionizing radiation | 6.32% | 5.39% | 0.05% | 27.91% | 60.33% |
| Ozone formation, human health | 7.57% | 15.48% | 2.11% | 16.77% | 58.07% |
| Fine particulate matter formation | 9.51% | 8.44% | 1.43% | 21.66% | 58.96% |
| Ozone formation, terrestrial ecosystems | 7.73% | 15.37% | 2.20% | 16.41% | 58.29% |
| Terrestrial acidification | 10.70% | 7.09% | 1.56% | 29.62% | 51.03% |
| Freshwater eutrophication | 2.54% | 6.45% | 0.13% | 18.00% | 72.88% |
| Marine eutrophication | 5.25% | 1.50% | 0.04% | 77.17% | 16.04% |
| Terrestrial ecotoxicity | 2.34% | 12.88% | 0.34% | 19.77% | 64.67% |
| Freshwater ecotoxicity | 1.94% | 16.06% | 0.20% | 20.40% | 61.40% |
| Marine ecotoxicity | 2.28% | 16.15% | 0.20% | 19.79% | 61.58% |
| Human carcinogenic toxicity | 3.00% | 10.37% | 0.98% | 12.51% | 73.13% |
| Human non-carcinogenic toxicity | 2.03% | 17.42% | 0.17% | 21.24% | 59.13% |
| Mineral resource scarcity | 2.68% | 8.65% | 0.16% | 39.51% | 49.00% |
| Fossil resource scarcity | 20.60% | 5.62% | 3.93% | 9.01% | 60.85% |

Field plastics are all plastics used in the production process. Agrochemicals include fertilizers and pesticides. Post-harvest includes washing, sorting, and packing.

Compared to organic production, the effect of input delivery on total GWP and other environmental impact categories is very small. Unlike organic production, fertilizer and other agrochemical use contribute substantially to all environmental impact categories. Again, post-harvest operations contribute the most to practically all environmental impact categories.

## 4. Discussion

This study compared the environmental impact of producing organic vs. conventional strawberries in the Central Valley region of the state of California using life cycle analysis. The functional unit of our life cycle analysis was a kg of strawberries produced, as well as a hectare of production. The system boundary of the study was from cradle to farmgate. The data on input requirements for each production system was obtained from production cost analyses published by UC Davis.

Our study showed that organically produced strawberries had a higher environmental impact in 16 out of 17 environmental categories than conventionally produced strawberries. A kg of organic strawberries has a carbon footprint 46% larger than a kg of conventionally produced strawberries. The environmental category with the largest impact difference

between the two production systems was marine eutrophication. Per kg of produce, organic strawberries were estimated to have 569% higher marine eutrophication impact, as measured in kg N eq, than conventional strawberries. Stratospheric ozone depletion, as measured in kg CFC11 eq per kg of produce, was the only environmental category where conventional strawberries performed better than organic strawberries (36% difference), when measured by functional unit.

When the environmental impact is measured in unit of area of production, the differences in environmental impact among all environmental categories becomes smaller. When measuring the carbon footprint per ha of production, organic strawberry production generates 14% more kg of $CO_2$ eq than conventional production. The environmental impact per unit of area of production is higher in organic production in 14 out of the 17 categories.

For both strawberry production systems, a large contributor to the carbon footprint per ha of production is post-harvest activities, which include product washing, sorting, and packing. Post-harvest activities in organic production contribute to about 40% of total carbon footprint per ha, while for conventional production, this value is 60%. Input transportation also contributes to a large percentage of total carbon footprint for organic production (33%) compared to conventional production (8%).

Some consumers are willing to pay a premium for organic produce in part because of the idea that organic crops are categorically superior to non-organic, conventional produce, in many environmental impact categories. This is not always the case. Organic fertilizers and inputs may have a very low environmental impact per unit of mass, and even if the volume is large, their effect is not very important. However, a consequence of using large volumes of compost, manure, and other organic inputs is the transportation impact of those inputs. Transportation, as we have shown, is a major contributor to GWP and most environmental impact categories. The effect of transportation is much smaller for conventional production.

As transportation technologies improve and are adopted by freight companies, the environmental impact on organic produce is expected to decrease substantially, if their net environmental benefits are larger than the impact of the current internal combustion engines. These coming years are part of a transition period for the adoption of hybrid and electric motors, and their environmental performance will be critical to their overall impact on the food industry, especially organic production.

## 5. Conclusions

We compared the environmental impact of strawberries production under organic and conventional production systems. We found that the environmental impact of organic strawberries is much higher in practically all impact categories compared to conventional production, under the assumption that they are grown in the same region in CA, and both systems follow the estimated input use published by UC Davis [12,13]. Due to the large volume of organic inputs needed in organic production of strawberries, transportation of inputs contributes significantly to total carbon footprint of organic strawberries. Understanding the critical processes that contribute significantly to the carbon footprint of strawberry production (or any other produce) allows us to efficiently respond and propose solutions to improve the environmental impact of the food system.

**Author Contributions:** Conceptualization, L.J.V.C.; Methodology, L.J.V.C.; Software, L.J.V.C.; Formal analysis, L.J.V.C.; Writing—original draft, L.J.V.C.; Writing—review & editing, M.I.G.; Funding acquisition, M.I.G. All authors have read and agreed to the published version of the manuscript.

**Funding:** This research received no external funding.

**Institutional Review Board Statement:** Not applicable.

**Informed Consent Statement:** Not applicable.

**Data Availability Statement:** Not applicable.

**Conflicts of Interest:** The authors declare no conflict of interest.

## Appendix A

**Table A1.** Inputs used to estimate the LCA of conventional strawberry production.

| Inputs | Quantity/ha | Unit |
|---|---|---|
| **Insecticide** | | |
| Savey 50 DF | 0.41 | kg |
| Dipel DF | 2.24 | kg |
| Success | 365.19 | mL |
| Acramite 50 WS | 2.24 | kg |
| Rimon 0.83 EC | 803.42 | mL |
| Malathion 8 | 2337.46 | mL |
| Beleaf 50 SG | 0.19 | kg |
| Danitol 2.4 EC | 1168.61 | mL |
| **Fungicide** | | |
| Captan 50 W | 17.90 | kg |
| Rally 40 W | 1.38 | kg |
| Pristine | 3.18 | kg |
| Quadris | 2629.36 | mL |
| Elevate 50 WDG | 3.36 | kg |
| Thiolux | 22.38 | kg |
| **Fertilizer** | | |
| Scotts 18-8-13 | 559.46 | kg |
| CAN 17 17-0-0 (N) | 391.62 | kg |
| **Materials** | | |
| T-Tape | 16,138.98 | m |
| Mulch 48" 1.25 mil | 8069.49 | m |
| Trays/Clamshells | 22,230.00 | units |
| **Water** | | |
| Water-Pumped | 69.85 | cm |
| **Machinery** | | |
| Fuel-Gas | 369.82 | L |
| Fuel-Diesel | 813.68 | L |

**Table A2.** Inputs used to estimate the LCA of organic strawberry production.

| Inputs | Quantity/ha | Unit |
|---|---|---|
| **Insecticide** | | |
| Vestis | 949.49 | mL |
| Dipel DF (Bt) | 4.48 | kg |
| **Fungicide** | | |
| Kumulus DF | 50.35 | kg |
| **Fertilizer** | | |
| Compost | 19.76 | ton |
| Feather Meal (13-0-0) | 1.24 | ton |
| Rice Bran | 12.35 | ton |
| Gypsum | 4.94 | ton |
| Biomin Calcium | 74.69 | L |
| Maxi-Crop | 22.38 | kg |
| Agrothrive LF | 1493.86 | L |
| True Organics 4-2-2 | 746.93 | L |
| **Materials** | | |
| T-Tape | 16,138.98 | m |
| Black Plastic Mulch | 6.79 | roll |
| Trays/Clamshells | 17,290.00 | units |
| **Water** | | |
| Water Pumped | 69.85 | cm ha |
| **Machinery** | | |
| Fuel-Gas | 654.87 | L |
| Fuel-Diesel | 1348.21 | L |

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
