# Peer review of "A Tale of Two Strawberries: Conventional and Organic Open-Field Production in California"

_sustainability, doi:10.3390/su151914363_

Round 1

Reviewer 1 Report

Dear  Author,

Considering the increasing importance of organic production recently, your work has scientific importance. However, there are scientific gaps, especially in materials and methods. I recommend that you take into consideration the revisions I have mentioned below.

-        In the introduction, information about the secondary matter content and nutritional value of the strawberry can be given.

-        There is no information about the material in the article. The method description is insufficient.

-        It is stated that the environmental impact of traditional and organic strawberries grown in the open field in California has been analyzed. How many years has this organic or classical farming been done in this field? What is the soil content of both cultivated fields?

-        Information about the region where the data is collected, the soil, the climate and the elemental contents of the classical or organic fertilizers used should be given. How many hectares of area were data collected from? The coordinates and other important ecological data of these areas can be given as a table. In data analysis, Dixon et al. (2019) (US Davis, 2022) following references such as fertilizer insecticides and fungicides amounts were determined. What method did these authors follow? Brief information should be given.

-        Actually there is no discussion in the article. The conlusion is not enough. The conclusion should be expanded by emphasizing the importance of the study.

Best regards,

Author Response

-        In the introduction, information about the secondary matter content and nutritional value of the strawberry can be given.

A: We included in the introduction information on the nutritional content and secondary metabolites between the two production systems.

-        There is no information about the material in the article. The method description is insufficient.

A: The material used in the LCA, that is, the life cycle inventory, was obtained from production cost estimates published by the extension program of UC Davis. We included the list of materials in the appendix as tables A1 and A2.

-        It is stated that the environmental impact of traditional and organic strawberries grown in the open field in California has been analyzed. How many years has this organic or classical farming been done in this field? What is the soil content of both cultivated fields?

A: The production cost estimates for both production systems are based on an average soil quality of the region. They are not calculated for a specific field.

-        Information about the region where the data is collected, the soil, the climate and the elemental contents of the classical or organic fertilizers used should be given. How many hectares of area were data collected from? The coordinates and other important ecological data of these areas can be given as a table. In data analysis, Dixon et al. (2019) (US Davis, 2022) following references such as fertilizer insecticides and fungicides amounts were determined. What method did these authors follow? Brief information should be given.

A: The intention of this study is not to compare specific farms, but two production systems of the same region. The data of the inputs used in both systems are estimated from cost analyses of expension programs of UC Davis and are estimated for a 30-acre operation of rented land in the Central Coast region, specifically in Santa Cruz and San Benito Counties. We clarify this in the Data section.

-        Actually there is no discussion in the article. The conlusion is not enough. The conclusion should be expanded by emphasizing the importance of the study.

A: We expanded the discussion and conclusion and slip them in its own section.

Reviewer 2 Report

Brief summary:

Thank you for giving me an opportunity to review the manuscript “A Tale of two strawberries: conventional and organic open field production in California”. The authors performed a life cycle analysis (LCA) on organic and conventionally produced strawberries in California, following input estimates from extension reports. The outcome of this study revealed that one of the main contributors to organic production's higher environmental impact is the effect of the transportation of compost, manure, and other organic inputs that are required in large volumes per hectare. In general, the manuscript is well structured and findings have significant implications for conventional and organic open field production of strawberries. I suggest this manuscript can be accepted after minor modifications. My specific comments are:

1.      Please add the varieties/species names of strawberries in the title.

2.      The abstract section should be reformulated while focusing on the major numerical findings of this study.

3.      Abstract: Write the full form of GWP.

4.      Do not use abbreviations in keywords.

5.      Please check if the reference style is as per MDPI style.

6.      Introduction: California is ranked 4th in which context (national or international?).

7.      Tables: Please correct subscripts for chemical names.

8.      The manuscript lacks definitions of several abbreviations.

9.      Please change hectare to “ha” in the entire manuscript.

10.   Figure 1: The text is too small please increase the font size.

11.   The discussion part should be rewritten as a separate section and not with the conclusion.

12.   References are okay.

Author Response

  1. Please add the varieties/species names of strawberries in the title.

 A: The study that estimated the list of materials used in the study do not specify a strawberry varietal but assumes that the cultural activities are typical of several varietals in the region. Thus, we cannot specify a varietal used in the study.

  1. The abstract section should be reformulated while focusing on the major numerical findings of this study.

A: We modified the abstract to focus on the major numerical findings.

  1. Abstract: Write the full form of GWP.

A: We spelled out GWP (Global Warming Potential)

  1. Do not use abbreviations in keywords.

            A: We eliminated abbreviations in keywords

  1. Please check if the reference style is as per MDPI style.

A: We changed the reference style to match that of MDPI.

  1. Introduction: California is ranked 4th in which context (national or international?).

A: We clarified that strawberry Is the 4th most valuable crop produced in California.

  1. Tables: Please correct subscripts for chemical names.

A: We modified the subscripts accordingly.

  1. The manuscript lacks definitions of several abbreviations.

A: Noted. The abbreviations have been defined throughout the paper.

  1. Please change hectare to “ha” in the entire manuscript.

A: Noted. We changed the abbreviation of hectare to ha.

  1. Figure 1: The text is too small please increase the font size.

A: We eliminated figure 1 after considering that the information presented is redundant.

  1. The discussion part should be rewritten as a separate section and not with the conclusion.

A: we separated the discussion and conclusion.

  1. References are okay.

            A: Thank you.

Round 2

Reviewer 1 Report

Dear Authors,

The corrections made by the authors are sufficient for me. The article is acceptable.

Best regards,